# Heatwaves and Mortality in Spain and Greece: A Comparative Analysis

**Lida Dimitriadou** [1,*] and **Christos Zerefos** [1,2,3,4]

1 Research Centre for Atmospheric Physics and Climatology, Academy of Athens, 106 79 Athina, Greece
2 Biomedical Research Foundation, Academy of Athens, 115 27 Athens, Greece
3 Navarino Environmental Observatory (N.E.O.), 240 01 Messinia, Greece
4 Mariolopoulos–Kanaginis Foundation for the Environmental Sciences, 106 75 Athens, Greece
* Correspondence: ldimitriadou@academyofathens.gr

**Abstract:** Background: Heatwaves have become a public health emergency raising great public health concerns. Correspondingly, due to climate change, the increase in extreme weather events, such as heatwaves, floods and hurricanes, is predicted through state-of-the-art models and robust statistical analyses. Since the Mediterranean basin is recognized as the most prominent global climate change hot spot, further scientific research on the future impacts of heatwaves on human mortality, as well as human health and well-being, in the Mediterranean region is critical. Methods: The aim of the present study is to compare the relationship between three different causes of mortality (i.e., cardiological, respiratory and cardiorespiratory) and temperature between two countries (Spain and Greece) and five cities (i.e., Madrid, Barcelona, Valencia, Attica and Thessaloniki). To compare the five cities under examination, a robust statistical methodological framework (i.e., Threshold Regression Analysis (TRA)) was applied so as to examine the critical value above which the mean temperature affects cardiological, respiratory and cardiorespiratory mortality. Results: Our results prove that the relationship between mortality and temperature is a nonlinear relationship shaping a U- or J-shaped curve, meaning that mortality is affected by temperature in a non-constant way, indicating that mortality increases during both low and high temperatures. Conclusions: By calculating different temperature thresholds for the five cities under examination, we show that different temperature thresholds are more accurate for different climatic conditions. Hence, climatic conditions should be taken into account when examining the impacts of temperature on human health.

**Keywords:** mortality; temperature; heatwaves; Spain; Greece

## 1. Introduction

The Mediterranean region is recognized as the most prominent global climate change hot spot. As a result, the Mediterranean basin warms more rapidly than other areas [1]. Furthermore, the region is particularly vulnerable to large-scale dynamics since it is located in a transitional zone between subtropical temperate and continental climate [2]. The Mediterranean region is defined as the land with a Mediterranean climate that is around the Mediterranean Sea [3]. Due to anthropogenic activities, the increase in greenhouse gases in the atmosphere, which is inextricably linked to the increase in mean temperature, is forecasted to augment and intensify extreme events, especially during the summer season (i.e., when heatwaves occur) [4]. Future models project rising concentrations of greenhouse gases more in the Mediterranean than in other land regions in relative terms [5]. Specifically, warming will be especially immense during the summer (approximately 50% greater than global warming) and for the northern regions of the basin (locally up to 100% greater than global warming) [5]. As a result, the decline of precipitation will impact all seasons in the central and southern Mediterranean areas, with a maximum reduction during the winter season [5]. Consequently, the heat stress risk is amplified in the Mediterranean region [6],

accelerating the burden and the vulnerability of the local population and affecting human health and well-being.

In meteorological terms, a heatwave is defined as a prolonged period of unusually hot weather [7]. Hitherto, there is not a universally accepted definition of heatwaves [8]. In essence, an event is considered as catastrophic or not depending on its intensity, frequency and duration [9]. Despite their detrimental impacts, definitions and measurements of heat waves are ambiguous and inconsistent, generally being endemic to only the group affected or the respective study reporting the analysis [10]. Much of the emerging literature echoes the increase in heatwave events with greater duration, intensity and frequency due to climate change with alarming impacts on human mortality and the environment [11–13]. This raises great public health concerns. Since heat is one of the deadliest weather-related phenomena [14], the future consequences of heatwave episodes are uncharted waters as warming trends are projected to intensify throughout the 21st century [1].

Several studies turn the spotlight on the relationship between sustained extreme heat (i.e., heatwaves) and human health and mortality. The temperature–mortality relationship is shown to be a U- or J-shaped in the relevant literature, indicating a nonlinear relationship underlying that mortality increases due to both high and low temperatures (see among others [15,16]). Population groups such as the elderly ([7,17]), pregnant women ([18,19]), children ([20,21]), people with a lower socioeconomic status ([22]) and people who live in urban areas ([23]) are identified to be more vulnerable to heat and its deleterious effects. The inclusion of the Mediterranean basin in the menu of heatwaves complicates the issue further. Specifically, parts of the Mediterranean region have already been affected by record-breaking temperatures, for example during summer 2003, which was the most severe and lethal heatwave Europe has ever experienced, and summer 2007, which was the hottest summer on record since 1891 [1].

Concerning summer 2003, critical regional and seasonal temperature variations were observed. More specifically, temperatures across the Mediterranean region and much of the northwestern part of the European continent were calculated to be above the 98th percentile of the 1961–1990 distribution averaged over the entire year [24]. During the heatwave of 2003, also known as the European heatwave, there was an estimated excess mortality of between 22,000 and 35,000 heat-related deaths across Europe [25]. However, according to other studies 45,000–55,000 deaths occurred throughout the heatwave [26]. Nonetheless, the spatial distribution of the event was not homogeneous [27]. For instance, in Spain the maximum daily temperature exceeded 40 °C across the country [24] with excess mortality potentially reaching 13,000 deaths [28]. Concerning summer 2007, Greece experienced the warmest summer of its instrumental history with record-breaking temperatures reaching up to 44.8 °C at the National Observatory of Athens [29]. Despite this, mortality was not at its peak, nor was it greater than during the summer of 1987 [29]. It is generally acknowledged that the persistency of a heatwave can be more important than its intensity [30]. Hence, the reason that summer 2007 resulted in a less destructive impact on human mortality than the summer of 1987 was its persistency as well as the low atmospheric humidity [29]. Moreover, since the public was more aware and early warning systems were applied after the European heatwave, mortality was not at its peak. While there is not a universally accepted definition of heatwaves, concerning Attica, a heatwave is defined as a period of at least 3 days when the mean temperature is higher than the 97.5th percentile [31]. Hence, heatwave definitions concern local climatic conditions, duration, intensity and frequency. Nonetheless, the inclusion of health effects due to the sustained extreme heat in any heatwave definition is vital.

As shown in the abovementioned literature review, different methodologies reach diverse conclusions. As established in the relevant literature, the Mediterranean basin is a climate change hot spot. Hence, in order to paint a more detailed picture and extract valuable conclusions, our main objectives with the present paper are (i) to compare two countries (i.e., Greece and Spain) and five different cities (i.e., Madrid, Barcelona, Valencia, Attica and Thessaloniki) that are located in the Mediterranean basin with different cli-

matic conditions concerning cardiological, respiratory and cardiorespiratory mortality during the summer season, (ii) to quantify a heatwave by using different temperature percentiles for each country and (iii) to compare how diverse climatic conditions define different percentiles that result in a heatwave. To do so, we used Threshold Regression Analysis (TRA) so as to introduce a temperature threshold in a nonlinear relationship (i.e., temperature and mortality). TRA is the optimal statistical tool to model nonlinearity using a diverse set of non-regular regression models that all depend on thresholds [32]. The reason for choosing Spain and Greece is that they are both located in the Mediterranean region, thus presenting a unique opportunity to compare different countries with diverse climatic conditions located in a significant hot spot area due to climate change. Due to a lack of data availability, it was not possible to include more countries and cities.

## 2. Materials and Methods

Our time series consist of daily frequency data that refer to the mean temperature and mortality for two different countries: Spain and Greece. Both countries are located in the Mediterranean basin, thus presenting a unique opportunity to compare them. Concerning Spain, we used the three largest cities (i.e., Madrid, Barcelona and Valencia) and the period under examination spans from 1 January 1980 to 31 December 2018 (13.514 days) on a daily frequency. The data concern cardiological, respiratory and cardiorespiratory mortality. Cardiorespiratory represents the cumulative burden of mortality from cardiological and respiratory mortality. The respective data were obtained from the Instituto Nacional de Estadística (https://www.ine.es/, accessed on 17 April 2023). Concerning Greece, we used the two largest cities: the Attica region (which contains the capital of Greece, namely Athens) and Thessaloniki, and the period under examination spans from 1 January 1992 to 2018 on a daily frequency (9.862 days) for Attica and from 1 January 1999 to 31 December 2018 on a daily frequency (7.305 days) for Thessaloniki. We collected the respective data concerning cardiological and respiratory mortality on a daily basis from the Hellenic Statistical Authority (https://www.statistics.gr/en/home/, accessed on 17 April 2023). Due to a lack of data availability, for Thessaloniki, we used only cardiological mortality. Furthermore, we used mortality data instead of morbidity since the mortality data are binary and consequently more easily collected and analyzed. Additionally, data collection concerning hospital admissions lacks consistency and may depend on each country's health system. Data on mean the temperature were obtained from the NASA Prediction of Worldwide Energy Resources (POWER). Table 1 displays the regions under examination with their corresponding time period as well as the cause of mortality.

**Table 1.** List of cities/regions under examination.

| Country | Region | Period | Cause of Mortality |
|---------|--------|--------|--------------------|
| Spain | Madrid | 1980–2018 | Cardiological, respiratory, and cardiorespiratory |
| | Barcelona | 1980–2018 | Cardiological, respiratory, and cardiorespiratory |
| | Valencia | 1980–2018 | Cardiological, respiratory, and cardiorespiratory |
| Greece | Attica | 1992–2018 | Cardiological, respiratory, and cardiorespiratory |
| | Thessaloniki | 1999–2018 | Cardiological |

Table 2 presents the cumulative mortality for the cities under examination (i.e., Madrid, Barcelona, Valencia, Attica and Thessaloniki). Cardiorespiratory mortality expresses the sum of cardiological, and respiratory mortality combined. As is shown by comparing the cities, cardiological and cardiorespiratory mortality is higher in Barcelona, while respiratory mortality is higher in Madrid.

**Table 2.** Cumulative amount of mortality for the cities under examination.

| Country | Region | Mortality | | |
|---|---|---|---|---|
| | | Cardiological | Respiratory | Cardiorespiratory |
| Spain | Madrid | 478.400 | 177.836 | 656.236 |
| | Barcelona | 533.782 | 151.985 | 685.767 |
| | Valencia | 293.306 | 79.379 | 372.685 |
| Greece | Attica | 279.325 | 88.438 | 367.763 |
| | Thessaloniki | 52.704 | - | - |

*Methodology*

In order to compare the two countries, Threshold Regression Analysis (TRA) was applied. TRA is the optimal statistical tool to model nonlinearity, using a diverse set of non-regular regression models that all depend on thresholds [32]. By introducing a threshold parameter, in the present study a temperature threshold, threshold regression models provide a simple but elegant and interpretable way to model certain kinds of nonlinear relationships between the outcome and a predictor [32]. In the present analysis, TRA was used to quantify the effect of heat (temperature) on mortality above a specific heat threshold. Threshold temperature (TT) is defined as the temperature beyond which mortality increases [33]. In essence, TT is the temperature above which mortality changes. The data series for the TRA consist of cardiological, respiratory and cardiorespiratory mortality (i.e., the sum of cardiological and respiratory mortality) and the mean temperature. The reason for choosing the mean and not the maximum temperature is that the mean temperature captures the nighttime temperatures more accurately (which may not be the highest temperature that contribute to the accumulative heat effect on the human body). The data concerning Spain span from 1 January 1980 to 31 December 2018 (13.514 days) on a daily frequency. The data concerning Greece span for Attica from 1 January 1992 to 2018 on a daily frequency (9.862 days) and for Thessaloniki from 1 January 1999 to 31 December 2018 on a daily frequency (7.305 days). TRA was quantified using EViews 10 statistical software. The formula for TRA is the following:

$$m_t = c_1 * T * I(T < b_1) + c_2 * T * I(b_1 < T < b_2) + c_3 * T * I(T < b_3) + c_4 + u_t \quad (1)$$

where $m_t$ is mortality, $c_1, c_2, c_3, c_4$ are parameters to be estimated, T is mean temperature, $b_1, b_2, b_3$ are the temperature thresholds, I($\bullet$) is an indicator factor which receives the value 1 if the condition in the parenthesis is true, while 0 otherwise and $u_t$ is the error term.

## 3. Results

Table 3 presents the descriptive statistics (i.e., minimum, mean, maximum mortality and standard deviation) for all causes of mortality and for the cities under examination. The numbers are expressed per 100,000 citizens so as to be more easily comparable. As is shown, maximum mortality is observed in cardiorespiratory mortality in Valencia (Spain) and in Attica (Greece).

Figure 1 illustrates different causes of mortality in Spain (i.e., cardiological, respiratory and cardiorespiratory), expressed per 100,000 citizens. As is shown, there is a downward trend for cardiological and cardiorespiratory mortality for all cities under examination, while there is an upward trend for respiratory mortality, indicating that respiratory mortality in Madrid, Barcelona and Valencia increases throughout the years. The visual representation of mortality was created using EViews 10 statistical software.

**Table 3.** Descriptive statistics concerning mortality in Spain and Greece.

| | Mortality | Minimum | Mean | Maximum | Std. Deviation |
|---|---|---|---|---|---|
| | | | Madrid | | |
| | Cardiological | 0.15 | 1.02 | 2.40 | 0.24 |
| | Respiratory | 0.00 | 0.37 | 1.60 | 0.21 |
| | Cardiorespiratory | 0.15 | 1.42 | 3.35 | 0.34 |
| | | | Barcelona | | |
| Spain | Cardiological | 0.80 | 2.31 | 5.43 | 0.57 |
| | Respiratory | 0.00 | 0.67 | 3.14 | 0.36 |
| | Cardiorespiratory | 1.05 | 2.98 | 7.96 | 0.77 |
| | | | Valencia | | |
| | Cardiological | 0.37 | 2.51 | 6.26 | 0.74 |
| | Respiratory | 0.00 | 0.69 | 3.32 | 0.40 |
| | Cardiorespiratory | 0.74 | 3.20 | 9.58 | 0.92 |
| | | | Attica | | |
| | Cardiological | 0.20 | 0.74 | 2.32 | 0.19 |
| Greece | Respiratory | 0.00 | 0.23 | 0.94 | 0.12 |
| | Cardiorespiratory | 0.26 | 0.97 | 2.69 | 0.25 |
| | | | Thessaloniki | | |
| | Cardiological | 0.00 | 0.38 | 1.12 | 0.15 |

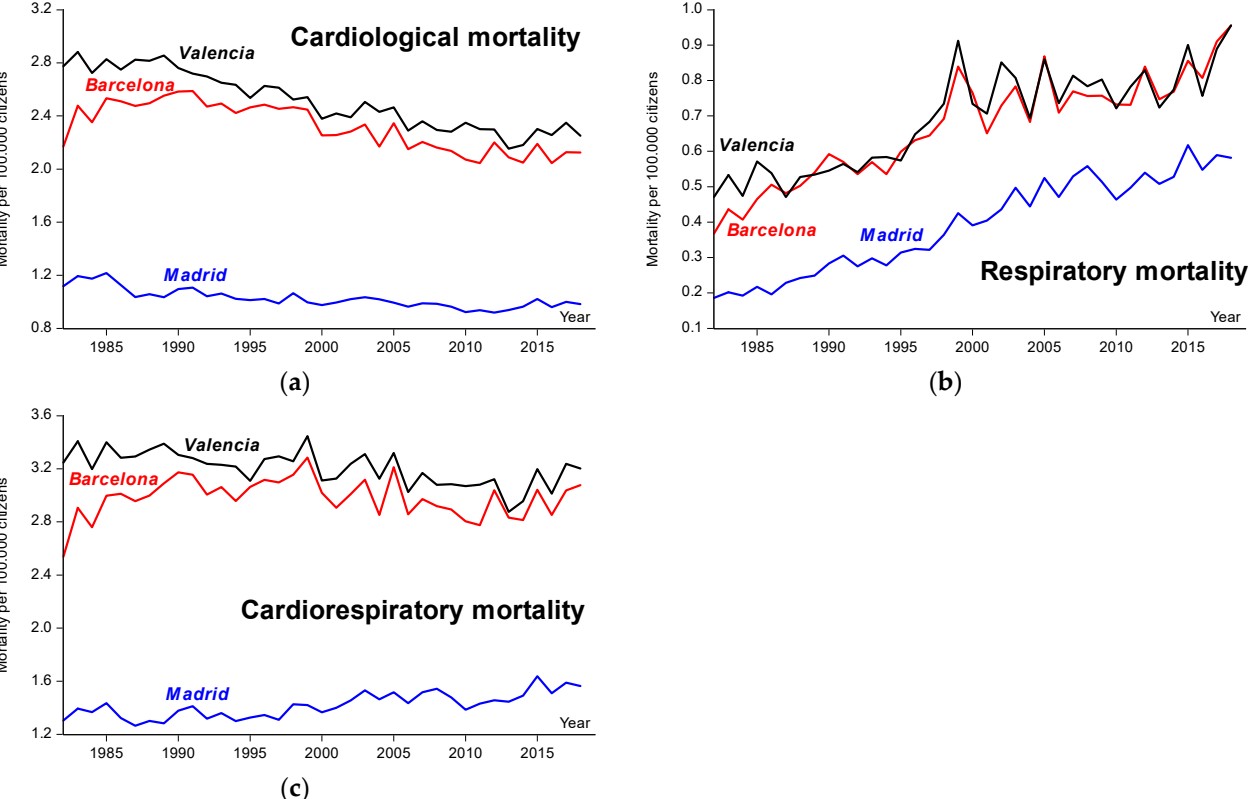

**Figure 1.** Mortality in Spain: (**a**) cardiological mortality, (**b**) respiratory mortality and (**c**) cardiorespiratory mortality.

Figure 2 presents the mortality graphs (i.e., cardiological, respiratory and cardiorespiratory mortality) for Greece (Attica and Thessaloniki) expressed per 100,000 citizens. As is shown, mortality for respiratory and cardiorespiratory increases throughout the years,

while cardiological mortality for both Attica and Thessaloniki remains stable. The visual representation of mortality was created using EViews 10 statistical software.

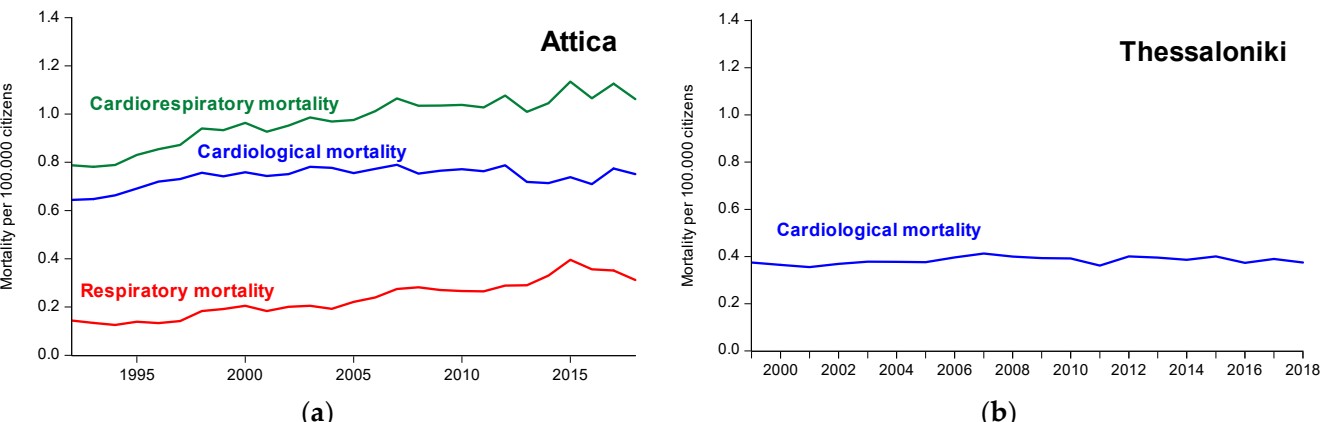

**Figure 2.** Mortality in Greece: (**a**) Athens and (**b**) Thessaloniki.

Figure 3 shows scatter plots for mortality expressed per 100,000 citizens and the mean temperature for Barcelona, Madrid and Valencia for cardiological, respiratory and cardiorespiratory mortality. The red line represents the Kernel fit line. Barcelona and Valencia show a distinct U-shaped relationship (especially for cardiological and cardiorespiratory mortality). In accordance with the relevant literature, the scatter plots indicate that the relationship between mortality and mean temperature is nonlinear, showing that mortality is affected by mean temperature in a non-constant way, meaning that mortality increases during both low and high temperatures. The visual representations of the scatter plots between mortality and mean temperature were created using the statistical software EViews 10.

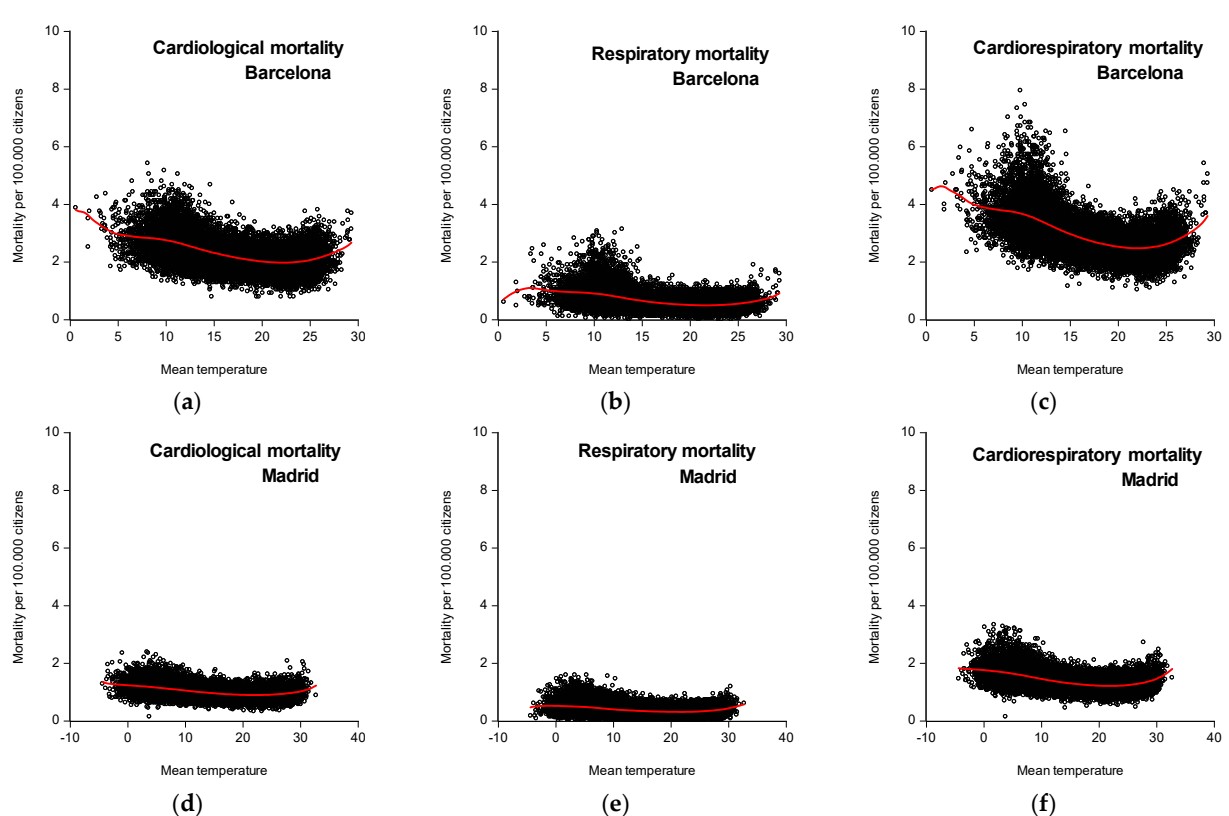

**Figure 3.** *Cont.*

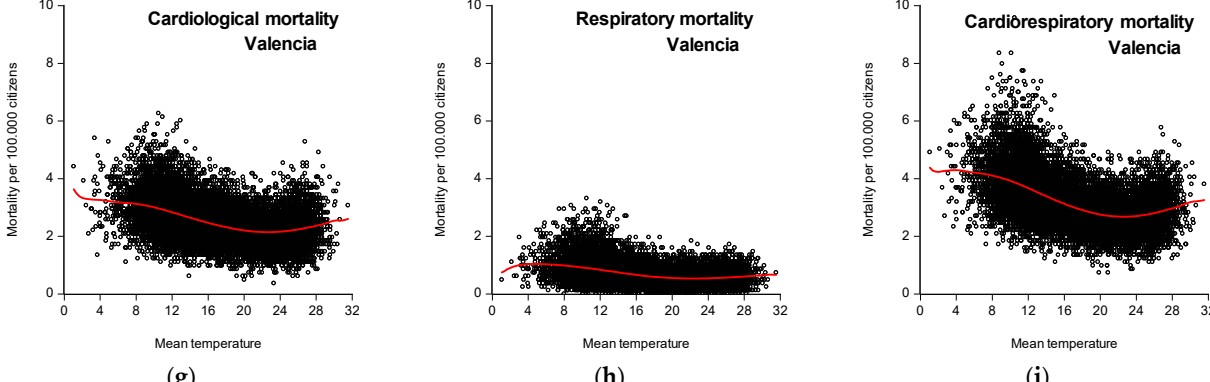

**Figure 3.** Scatter plots for Spain for all causes of mortality under examination and the mean temperature: (**a**) cardiological mortality for Barcelona, (**b**) respiratory mortality for Barcelona, (**c**) cardiorespiratory mortality for Barcelona, (**d**) cardiological mortality for Madrid, (**e**) respiratory mortality for Madrid, (**f**) cardiorespiratory mortality for Madrid, (**g**) cardiological mortality for Valencia, (**h**) respiratory mortality for Valencia, and (**i**) cardiorespiratory mortality for Valencia.

In line with Figure 3, Figure 4 presents the scatter plots for the causes of mortality under examination and the mean temperature for Greece. The red line represents the Kernel fit line. In line with the above graphs and the relevant literature, a U-shaped relationship is shown, specifically for cardiological and cardiorespiratory mortality, proving that the mean temperature affects mortality in a non-constant way, meaning that mortality increases during both low and high temperatures. The visual representations of the scatter plots between mortality and mean temperature were created using the statistical software EViews 10.

Tables 4 and 5 present the mean and maximum temperature percentiles under examination (i.e., the 95th, 97.5th and 99th) for two different heatwave event durations (a duration equal to or greater than 2 and 3 days, respectively), as well as the days that the respective temperatures exceed the percentile value ("*Days*"). The events ("*Events*") are the days where the temperature exceeds the percentile value with a duration equal to or greater than 2 and 3 days.

**Table 4.** Temperature percentiles, number of days and number of events when the heatwave duration is equal to or greater than 2 days.

| Percentiles | 95th | Days | Events | 97.5th | Days | Events | 99th | Days | Events |
|---|---|---|---|---|---|---|---|---|---|
| | | | | Madrid | | | | | |
| Mean T | 29.11 | 169 | 34 | 29.85 | 83 | 17 | 30.47 | 33 | 7 |
| Max T | 37.86 | 168 | 35 | 38.56 | 84 | 19 | 39.42 | 33 | 8 |
| | | | | Barcelona | | | | | |
| Mean T | 26.41 | 169 | 27 | 26.86 | 83 | 13 | 27.52 | 33 | 6 |
| Max T | 29.06 | 169 | 34 | 29.66 | 84 | 14 | 30.41 | 33 | 6 |
| | | | | Valencia | | | | | |
| Mean T | 27.92 | 168 | 31 | 28.47 | 83 | 17 | 29.06 | 33 | 6 |
| Max T | 33.07 | 170 | 29 | 33.77 | 84 | 12 | 34.87 | 33 | 5 |
| | | | | Attica | | | | | |
| Mean T | 30.13 | 123 | 29 | 30.92 | 61 | 16 | 31.86 | 24 | 5 |
| Max T | 36.55 | 123 | 26 | 37.62 | 60 | 9 | 39.29 | 24 | 4 |
| | | | | Thessaloniki | | | | | |
| Mean T | 30.15 | 90 | 18 | 30.92 | 45 | 12 | 31.94 | 17 | 5 |
| Max T | 37.05 | 90 | 20 | 37.87 | 45 | 12 | 39.42 | 18 | 6 |

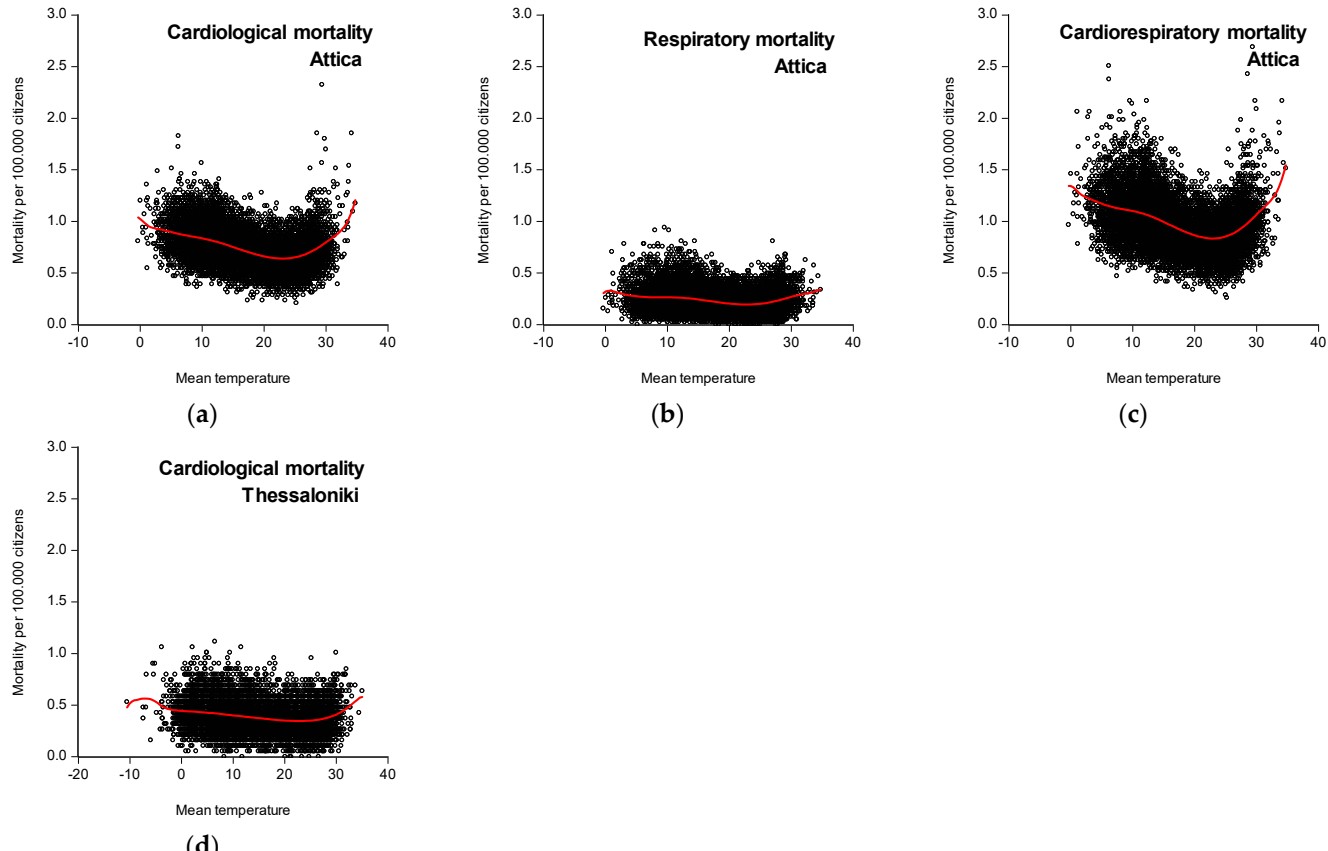

**Figure 4.** Scatter plots for Greece for all causes of mortality under examination and the mean temperature: (**a**) cardiological mortality for Attica, (**b**) respiratory mortality for Attica, (**c**) cardiorespiratory mortality for Attica, and (**d**) cardiological mortality for Thessaloniki.

**Table 5.** Temperature percentiles, number of days and number of events when the heatwave duration is equal to or greater than 3 days.

| Percentiles | 95th | Days | Events | 97.5th | Days | Events | 99th | Days | Events |
|---|---|---|---|---|---|---|---|---|---|
| Madrid | | | | | | | | | |
| Mean T | 29.11 | 169 | 20 | 29.85 | 83 | 10 | 30.47 | 33 | 2 |
| Max T | 37.86 | 168 | 17 | 38.56 | 84 | 7 | 39.42 | 33 | 2 |
| Barcelona | | | | | | | | | |
| Mean T | 26.41 | 169 | 17 | 26.86 | 83 | 9 | 27.52 | 33 | 5 |
| Max T | 29.06 | 169 | 18 | 29.66 | 84 | 7 | 30.41 | 33 | 3 |
| Valencia | | | | | | | | | |
| Mean T | 27.92 | 168 | 13 | 28.47 | 83 | 5 | 29.06 | 33 | 1 |
| Max T | 33.07 | 170 | 6 | 33.77 | 84 | 4 | 34.87 | 33 | 3 |
| Attica | | | | | | | | | |
| Mean T | 30.13 | 123 | 14 | 30.92 | 61 | 8 | 31.86 | 24 | 3 |
| Max T | 36.55 | 123 | 9 | 37.62 | 60 | 5 | 39.29 | 24 | 2 |
| Thessaloniki | | | | | | | | | |
| Mean T | 30.15 | 90 | 12 | 30.92 | 45 | 6 | 31.94 | 17 | 1 |
| Max T | 37.05 | 90 | 13 | 37.87 | 45 | 5 | 39.42 | 18 | 2 |

By comparing the different cities and different percentiles, it is shown that (i) by defining different temperature percentiles, the number of days and events is altered, (ii) Greece

has higher temperature percentile values, which means that climatic conditions that create a heatwave in Greece do not define a heatwave in Spain and vice versa and consequently (iii) heatwaves for Greece are characterized by higher ambient temperatures compared to Spain.

*Threshold Regression Analysis*

In the previous sections, examination and comparison of the two countries (Spain and Greece) occurred. In the present section, Threshold Regression Analysis (TRA) is conducted as a robust statistical analysis so as to examine the critical value above which the mean temperature affects cardiological, respiratory and cardiorespiratory mortality for the five cities under examination. Table 6 demonstrates the temperature thresholds for each city and the different causes of mortality. The bold numbers indicate that the values are statistically significant at all levels of significance (i.e., 10%, 5% and 1%). As demonstrated, different temperature thresholds apply for different cities. The levels of significance along with the associated sign confirm whether the relationship between mortality and temperature is a U- or J-shaped relationship (shown in the scatter plots in the previous section). If the temperature thresholds are not statistically significant, there is no change in the slope of the curve. The generated thresholds for all countries under examination were calculated using EViews 10 statistical software. Concerning cardiological mortality, Thessaloniki demonstrated the lowest value on the lower threshold and the highest value on the upper threshold. Concerning respiratory mortality, Madrid demonstrated the lowest value on the lower threshold, while Attica presented the highest value on the upper threshold. Lastly, concerning cardiorespiratory mortality, Attica demonstrated the lowest value on the lower threshold, while Madrid presented the highest value on the upper threshold. In conclusion, by defining lower and upper thresholds for different cities with diverse climatic conditions, various conclusions can be drawn.

**Table 6.** Temperature thresholds for the different cities.

| | Country | Region/Cities | Longitude | Latitude | Lower Threshold | Upper Threshold |
|---|---|---|---|---|---|---|
| **Cardiological mortality** | **Spain** | Madrid | −3.70379 | 40.41678 | **17.61** | **27.24** |
| | | Barcelona | 2.15401 | 41.39021 | **12.98** | **22.13** |
| | | Valencia | −0.375000 | 39.46667 | **12.45** | **22.26** |
| | **Greece** | Attica | 23.72754 | 37.98381 | **13.55** | **24.53** |
| | | Thessaloniki | 22.94741 | 40.62927 | **7.61** | **27.48** |
| **Respiratory mortality** | **Spain** | Madrid | −3.70379 | 40.41678 | **7.24** | **21.54** |
| | | Barcelona | 2.15401 | 41.39021 | **12.24** | **20.44** |
| | | Valencia | −0.375000 | 39.46667 | **12.69** | **19.65** |
| | **Greece** | Attica | 23.72754 | 37.98381 | **15.00** | **24.44** |
| **Cardiorespiratory mortality** | **Spain** | Madrid | −3.70379 | 40.41678 | **17.61** | **26.98** |
| | | Barcelona | 2.15401 | 41.39021 | **12.98** | **22.12** |
| | | Valencia | −0.375000 | 39.46667 | **12.55** | **21.90** |
| | **Greece** | Attica | 23.72754 | 37.98381 | **9.76** | **24.23** |

## 4. Conclusions

As extreme events increase due to climate change, heatwaves are projected to become more frequent, with greater intensity, timing and duration. Consequently, early warning systems, preventive and adaptive measures and public awareness are required so as to prevent future mortality.

The aim of the present study was to compare two different countries located in the Mediterranean region, i.e., Spain and Greece, and five different cities, i.e., Madrid, Barcelona, Valencia, Attica and Thessaloniki. The reason that Spain and Greece were chosen is because they both belong in the Mediterranean basin, which is characterized as a global climate change hot spot. Hence, the spotlight was turned on them so as to compare countries that are located in a climate change hot spot area but simultaneously have different climatic conditions. The data under examination concern cardiological, respiratory and cardiorespiratory mortality and the mean temperature so as to capture the effect of temperature on mortality. After visual representation of mortality and the mean temperature (i.e., scatter plots), we confirmed that the relationship between temperature and mortality is nonlinear, thus agreeing with the results of the existing literature (see among others [34–37]). Furthermore, visual representation of mortality shows a downward trend for cardiological and cardiorespiratory mortality and an upward trend for respiratory mortality for Spain, while for Greece, respiratory and cardiorespiratory mortality increase throughout the years, and the trend remains approximately stable for cardiological mortality. Our results agree with the relevant literature since the scatter plots reveal a U-shaped relationship between mortality and temperature implying that temperature affects mortality in a nonlinear way [38,39]. In light of nonlinearity, the temperature percentiles were quantified so as to observe how different percentiles may be defined by different temperature thresholds for the five cities under examination.

Different durations (i.e., a duration equal to or greater than 2 and 3 days) of the heatwave events were applied as well. It is critical to observe that by defining different temperature percentiles, diverse heatwave definitions are created. Specifically, by comparing the different cities and the different percentiles, it was shown that (i) by defining different temperature percentiles, the number of days and events was altered, (ii) Greece has higher temperature percentile values, which means that climatic conditions that create a heatwave in Greece do not define a heatwave in Spain and vice versa, and consequently (iii) heatwaves in Greece are characterized by higher ambient temperatures compared to Spain.

By quantifying lower and upper thresholds for the five cities, we extract the following conclusions: concerning cardiological mortality, Thessaloniki demonstrates the lowest value on the lower threshold and the highest value on the upper threshold; concerning respiratory mortality, Madrid demonstrates the lowest value on the lower threshold, while Attica presents the highest value on the upper threshold; and lastly, concerning cardiorespiratory mortality, Attica demonstrates the lowest value on the lower threshold, while Madrid presents the highest value on the upper threshold. In conclusion, by defining lower and upper thresholds for different cities with diverse climatic conditions, various conclusions can be drawn. This finding is in agreement with the relevant literature that suggests that by defining different percentiles and comparing several countries with diverse climatic conditions, different conclusions can be extracted [6,21,35,36]. Moreover, definitions based on assumptions about temperature may lead to wrong conclusions, since similar temperatures can have different impacts depending on the duration, intensity and frequency of an event, as well as the acclimatization status of the population from one country to another [8,10,40]. Consequently, different preventive and control measures are to be decided. Furthermore, temperature thresholds were quantified (minimum and maximum thresholds) for the cities under consideration so as to observe how temperature affects different climatic conditions. To address all of the above issues that emerge due to increased anthropogenic activities that lead to intensification of extreme events, early warning systems and preventive measures should be applied on a national scale since different climatic conditions may consist of a heatwave in one country but not in another.

**Author Contributions:** Conceptualization, L.D. and C.Z.; methodology, L.D.; software, L.D.; validation, L.D. and C.Z.; formal analysis, L.D.; investigation, L.D.; resources L.D.; data curation, L.D.; writing—original draft preparation, L.D.; writing—review and editing, C.Z.; visualization, L.D.;

supervision, C.Z.; project administration, C.Z.; funding acquisition, C.Z. All authors have read and agreed to the published version of the manuscript.

**Funding:** This research received no external funding.

**Institutional Review Board Statement:** Not applicable.

**Informed Consent Statement:** Not applicable.

**Data Availability Statement:** Not applicable.

**Acknowledgments:** The study was financially supported by the Mariolopoulos-Kanaginis Foundation for the Environmental Sciences.

**Conflicts of Interest:** The authors declare no conflict of interest.

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
