# Peer review of "Heatwaves and Mortality in Spain and Greece: A Comparative Analysis"

_atmosphere, doi:10.3390/atmos14050766_

Round 1
Reviewer 1 Report
The authors carried out systematic studies on examining the relationship between three different causes of mortality (i.e., cardiological, respiratory, and cardiorespiratory) and temperature for five cities in two countries. The topic is interesting, and I believe it is important in the related area. The approach could be significant research on exploring the critical value above which mean temperature affects cardiological, respiratory, and cardiorespiratory mortality across developed and developing country cities. Therefore, the manuscript is recommended for publishing in Int. J. Environ. Res. Public Health subjected to minor revisions. More details are as follows:
1. Please rewrite the sentence “Moreover, Threshold Regression Analysis (TRA) is applied so as to examine the critical value above which mean temperature affects cardiological, respiratory and cardiorespiratory mortality for the 5 cities under examination” in the Abstract.
2. The advantage of the adopted approach should be strengthened in the introduction. The literature review part should be rewritten. The pros and cons of existing studies are not well presented.
3. The citation numbers should be superscripted.
4. Some formulas are printed in different styles, please make them in a unifying form.
5. The advantage of the adopted approach should be strengthened in the introduction.
6. Some tables should be revised to improve their quality and keep format consistency in presenting them. For example, Tables 4 and 5 are formatted rather odd, please reform them.
7. The paper suffers from some language problems and structural problems that make it difficult to assess the technical merit of the research. Recommend that the paper be polished very carefully.
Author Response
Please see also the attachment
Point 1: Please rewrite the sentence “Moreover, Threshold Regression Analysis (TRA) is applied so as to examine the critical value above which mean temperature affects cardiological, respiratory and cardiorespiratory mortality for the 5 cities under examination” in the Abstract.
Response 1: We thank the reviewer for this essential comment. We have rewritten this sentence.
Point 2: The advantage of the adopted approach should be strengthened in the introduction. The literature review part should be rewritten. The pros and cons of existing studies are not well presented.
Response 2: We thank the reviewer for this critical comment. We have corrected and strengthened the introduction as well as the literature review sections.
Point 3: The citation numbers should be superscripted.
Response 3: Thank you for the comment. However, according to the submission guidelines of the journal there is no indication that the citation numbers should be superscripted. Specifically, the guidelines state:
“References must be numbered in order of appearance in the text (including citations in tables and legends) and listed individually at the end of the manuscript. We recommend preparing the references with a bibliography software package, such as EndNote, ReferenceManager or Zotero to avoid typing mistakes and duplicated references. Include the digital object identifier (DOI) for all references where available. In the text, reference numbers should be placed in square brackets [ ] and placed before the punctuation; for example [1], [1–3] or [1,3]. For embedded citations in the text with pagination, use both parentheses and brackets to indicate the reference number and page numbers; for example”.
Point 4: Some formulas are printed in different styles, please make them in a unifying form.
Response 4: We made them in a unifying form. Thank you
Point 5: The advantage of the adopted approach should be strengthened in the introduction.
Response 5: We have strengthened the adopted approach in the introduction.
Point 6: Some tables should be revised to improve their quality and keep format consistency in presenting them. For example, Tables 4 and 5 are formatted rather odd, please reform them.
Response 6: We have formatted all the tables in a consistent way.
Point 7: The paper suffers from some language problems and structural problems that make it difficult to assess the technical merit of the research. Recommend that the paper be polished very carefully.
Response 7: We have revised the whole paper. Thank you for the insightful comment.

Reviewer 2 Report
The paper is on the relationship between heatwaves and mortality. In terms of the significance of heatwaves and mortality among public health topic, the paper meets the journal scope. There are some suggestions that need to be acknowledged.
1. The third sentence in the abstract is not meaningful. Maybe you should empasize that the study on Mediterranean basin is critical, rather than the study is critical.
2. Please state the contributions of this paper in the Introduction, based on a comparison with existing literature.
3. Please highlight why you aim to compare the two countries.
4. Ecourage more comparison between the two countries. I did not see enough discussion on the difference between countries in the current version.
5. Some specific results, such as numbers, should be included in the conclusion.
Author Response
Please see also the attachment
Point 1: The third sentence in the abstract is not meaningful. Maybe you should emphasize that the study on Mediterranean basin is critical, rather than the study is critical.
Response 1: Thank you for the insightful comment. We have emphasized that.
Point 2: Please state the contributions of this paper in the Introduction, based on a comparison with existing literature.
Response 2: We have stated the contributions of our paper. The following paragraph has been added.
As shown in the abovementioned literature review, different methodologies reach diverse conclusions. As established in the relevant literature, the Mediterranean basin is a climate change hotspot. Hence, in order to paint a more detailed picture and extract valuable conclusions our main objectives with the present paper are (i) to compare two countries (i.e., Greece and Spain) and five different cities (i.e., Madrid, Barcelona, Valencia, Attica and Thessaloniki) that are located in the Mediterranean basin with different climatic conditions concerning cardiological, respiratory and cardiorespiratory mortality during the summer season, (ii) to quantify a heatwave by using different temperature percentiles for each country, and (iii) to compare how different percentiles define different climatic conditions that result in a heatwave. To do so, we use Threshold Regression Analysis (TRA) so as to introduce a temperature threshold in a nonlinear relationship (i.e., temperature and mortality). TRA is the optimal statistical tool to model nonlinearity using a diverse set of non-regular regression models that all depend on thresholds [32]. The reason of choosing Spain and Greece is that they are both located in the Mediterranean region presenting a unique opportunity to compare different countries with diverse climatic conditions located in the most hotspot area due to climate change. Due to lack of data availability, it was not possible to include more countries and cities.
Point 3: Please highlight why you aim to compare the two countries.
Response 3: Thank you for the critical comment. We have highlighted both in the abstract, introduction and conclusions that we compare these two countries since they both belong in the Mediterranean region and we would like to examine what happens in the same regions where different climatic conditions are applied.
Point 4: Ecourage more comparison between the two countries. I did not see enough discussion on the difference between countries in the current version.
Response 4: Thank you very much for the comment. We have added comparison between the two countries.
Point 5: Some specific results, such as numbers, should be included in the conclusion.
Response 5: We have included our results in the conclusions.
